# Computing Elastic Moduli of Igneous Rocks Using Modal Composition and Effective Medium Theory

**Saeed Aligholi * and Manoj Khandelwal**

Institute of Innovation, Science and Sustainability, Federation University Australia, Ballarat, VIC 3350, Australia
* Correspondence: s.aligholi@federation.edu.au

**Abstract:** Elastic constants of rock materials are the basic parameters required for modeling the response of rock materials under mechanical loads. Experimental tests for determining these properties are expensive, time-consuming and suffer from a high uncertainty due to both experimental limitations and the heterogeneous nature of rock materials. To avoid such experimental difficulties, in this paper a method is suggested for determining elastic constants of rock materials by determining their porosity and modal composition and employing effective medium theory. The Voigt–Reuss–Hill average is used to determine effective elastic constants of the studied igneous rocks according to the elastic moduli of their mineral constituents. Then, the effect of porosity has been taken into account by considering rock as a two-phase material, and the Kuster–Toksoz formulation is used for providing a close estimation of different moduli. The solutions are provided for different isotropic igneous rocks. This sustainable method avoids destructive tests and the usage of energy for performing time-consuming and expensive tests and requires simple equipment.

**Keywords:** elasticity; Voigt–Reuss–Hill average; Kuster–Toksoz formulation; mineralogy

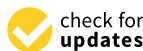



## 1. Introduction

There are various direct and indirect methods to estimate different moduli of elasticity. The Young modulus ($E$) and Poisson ratio ($v$) of rock materials, for instance, can be directly computed from uniaxial comparative strength (UCS) tests [1,2]. The computed values following these common methods are based on the following assumptions: isotropy, homogeneity and linear elasticity. The rock materials, however, are neither homogeneous nor linear elastic. Therefore, repeating several expensive and time-consuming standard UCS tests is required to estimate the average elastic moduli of the rocks with the assumption of isotropy. If the rock is not isotropic, then this method is not acceptable. It is notable that igneous rock materials can be considered isotropic, but sedimentary and metamorphic rocks are mainly transversely isotropic. For modeling the deformation of transversely isotropic rocks, the estimation of five different elastic moduli is required [3].

With the assumption of isotropic deformation by having any two elastic moduli such as $E$ and $v$, the others can be determined using some relationships that are basically derived from Hooke's law. The shear modulus ($G$) and bulk modulus ($K$) can also be determined experimentally. However, direct methods are costly and time-consuming. Furthermore, sometimes it is nearly impossible to measure such parameters using direct laboratory or in situ tests because of experimental limitations. Elastic constants are of great importance to material and structural engineers, and various indirect methods have been developed to estimate different elastic moduli. For example, compressional and shear wave velocities have been used for estimating the elastic constants of subsurface rocks [4]. In this study, various elastic constants of different heterogeneous igneous rocks that can be considered isotropic are determined by employing the effective medium theory. The adopted methodology requires no sophisticated equipment and can provide a good estimation of the elastic constants of rocks for modeling deformation and failure analyses of rock structures.

## 2. Materials and Methods

A wide range of Iranian volcanic and plutonic igneous rocks are studied. Table 1 summarizes the conducted modal analysis of the studied rocks. The modal composition of the studied rocks is determined by means of digital optical microscopy [5]. The porosity of the studied rocks is determined according to [6].

**Table 1.** Modal analysis and rock names of the studied rocks (the mineral constituents are given in percentage).

| Rock Code | Qtz | Pl | Afs | Bt | Ms | Am | Chl | Cpx | Opx | Ol | Grt | OM | Ep | Gl | AM | Rock Name * |
|---|---|---|---|---|---|---|---|---|---|---|---|---|---|---|---|---|
| R1 | 28 | 35 | 20 | 17 | - | - | - | - | - | - | - | - | - | - | Zrn | Micro-monzogranite |
| R2 | 38 | 24 | 28 | 3 | 7 | - | - | - | - | - | - | - | - | - | - | Monzogranite |
| R3 | 31 | 21 | 34 | 4 | 9 | - | 1 | - | - | - | - | - | - | - | - | Monzogranite |
| R4 | - | 41 | - | - | - | 5 | - | 8 | 5 | - | 2 | 2 | 1 | 36 | - | Hyalo-basaltic andesite |
| R5 | 32 | 34 | 27 | 4 | - | 2 | - | - | - | - | - | 1 | - | - | - | Monzogranite |
| R6 | - | 47 | - | - | - | 4 | - | 19 | 5 | 9 | - | 13 | 3 | - | - | Basalt |
| R7 | 37 | 22 | 31 | 7 | - | 1 | 2 | - | - | - | - | - | - | - | Zrn | Monzogranite |
| R8 | 37 | 18 | 38 | 5 | - | 1 | 1 | - | - | - | - | - | - | - | Zrn | Syenogranite |
| R9 | 41 | 14 | 40 | 2 | - | 1 | 2 | - | - | - | - | - | - | - | Zrn | Syenogranite |
| R10 | 26 | 17 | 48 | 6 | - | - | 1 | - | - | - | - | - | 2 | - | - | Syenogranite |
| R11 | 12 | 34 | 7 | 9 | - | 3 | - | - | - | - | - | 3 | - | 32 | - | Hyalo-dacite |
| R12 | 29 | 38 | 22 | 4 | - | 3 | 2 | - | - | - | 1 | 1 | - | - | - | Monzogranite |
| R13 | 14 | 46 | 16 | - | - | 12 | 9 | - | - | - | - | 3 | - | - | Zrn | Quartz Monzodiorite |
| R14 | 12 | 59 | 3 | - | - | 7 | 9 | 4 | - | - | - | 6 | - | - | - | Andesite |
| R15 | 1 | 64 | 1 | 3 | - | 23 | 3 | - | - | - | 1 | 4 | - | - | Spn | Diorite |
| R16 | 4 | 63 | 6 | - | - | 14 | 8 | - | - | - | - | 5 | - | - | - | Andesite |
| R17 | 28 | 29 | 35 | 5 | - | - | 3 | - | - | - | - | - | - | - | - | Monzogranite |
| R18 | - | 48 | - | 4 | - | 3 | - | 16 | 6 | - | 9 | 6 | 8 | - | - | Gabbro |
| R19 | - | 36 | - | 5 | - | 7 | - | 18 | 11 | 14 | - | 2 | 7 | - | - | Gabbro |
| R20 | - | 57 | 2 | 4 | - | 6 | - | - | 11 | 8 | 1 | 5 | 6 | - | Zrn | Diorite |
| R21 | - | 58 | 3 | 4 | - | 4 | - | 13 | 3 | 7 | 1 | 4 | 3 | - | - | Diorite |
| R22 | 13 | 33 | 41 | 1 | - | 9 | - | - | - | - | - | 3 | - | - | - | Quartz Monzonite |
| R23 | 7 | 40 | 44 | 1 | - | 6 | - | - | - | - | - | 2 | - | - | - | Quartz Monzonite |
| R24 | 12 | 24 | 56 | 2 | - | 4 | - | - | - | - | - | 2 | - | - | - | Quartz Syenite |
| R25 | 3 | 69 | 7 | 4 | - | 14 | - | - | - | - | - | 3 | - | - | - | Andesite |
| R26 | 27 | 41 | 19 | 6 | - | 4 | 1 | - | - | - | - | 2 | - | - | Spn | Granodiorite |
| R27 | 29 | 45 | 9 | 7 | - | 6 | 2 | - | - | - | 1 | 1 | - | - | Spn | Granodiorite |
| R28 | - | 59 | - | 8 | - | 5 | - | 12 | 4 | - | - | 2 | 6 | 4 | - | Micro-gabbro |

* According to optical microscopy studies (Streckeisen, 1976). Qtz: quartz; Pl: plagioclase; Afs: alkali feldspar; Bt: biotite; Ms: muscovite; Am: amphibole; Chl: chlorite; Cpx: clino pyroxene; Opx: orthopyroxene; Ol: olivine; Grt: garnet; OM: OPAC minerals; Ep: epidote; Gl: glass; AM: accessory minerals; Zrn: zircon; Spn: sphene.

Elastic constants of different mineral phases have been experimentally determined and compiled by different researchers. In this study, the values compiled by Bass [7] and Mavko et al. [8] are used. Table 2 summarizes the elastic constants of the mineral constituents of the studied igneous rocks. The elastic moduli of different mineral constituents of each rock are used to determine its effective elastic properties by employing the Voigt–Reuss–Hill average [9].

**Table 2.** Bulk and shear moduli of constituents of the studied igneous rocks.

| Elastic Modulus | Qtz | Pl | Afs | Bt | Ms | Am | Chl | Cpx | Opx | Ol | Grt | OM | Ep | Gl (Sio$_2$) | Gl (Andesite) | Gl (Basalt) |
|---|---|---|---|---|---|---|---|---|---|---|---|---|---|---|---|---|
| K | 37.7 | 70.6 | 56.2 | 49.9 | 58.2 | 90.2 | 81.0 | 104.5 | 104.4 | 131.8 | 159.4 | 142.6 | 106.2 | 36.5 | 52.5 | 62.9 |
| G | 44.4 | 34.3 | 28.4 | 27.1 | 35.3 | 46.2 | 27.0 | 62.0 | 63.9 | 65.9 | 90.4 | 114.5 | 61.2 | 31.2 | 33.6 | 36.5 |

If optical microscopy is used, the values for different mineral phases must be chosen based on some petrological facts and engineering judgements. If a rock is igneous, its rock-forming minerals are different from a metamorphic rock. The common clino-pyroxenes, for instance, in igneous rocks are augite and diopside, so in this study, the average bulk and shear moduli of these two minerals are considered for the studied igneous rocks. Some mineral groups are solid solutions and have two end members. The olivine mineral group, for example, has two end members, namely forsterite ($Mg_2SiO_4$) and fayalite ($Fe_2SiO_4$), and

most olivine minerals fall somewhere between them. In this study the average modulus of pure forsterite and fayalite is used for the olivine mineral group; the same is true for other solid solutions such as the plagioclase and alkali feldspar mineral groups. In general, the authors recommend the reported average modulus for igneous rocks (Table 2). For better estimations, one can use more accurate mineral identification methods, such as using XRD.

### 2.1. Determining Modal Composition

For the successful application of this method, determining the mineral composition of the rock should be obtained from a representative sample. Selecting a representative thin section and determining the modal composition of the studied rocks are explained in [10]. Notably, automatic mineral identification schemes [11–13] can be employed for fast and reliable determination of the modal composition of rock materials as the basic requirement of this method.

### 2.2. Hooke's Law

The deformation of a material under an applied load is controlled by its stiffness matrix. In other words, the elastic constants of material are making a relationship between the infinitesimal strain ($\varepsilon_{kl}$) and Cauchy stress ($\sigma_{ij}$), which are second-order tensors. Therefore, the deformation of a material can be determined by a fourth order tensor $C_{ijkl}$:

$$\sigma_{ij} = C_{ijkl}\varepsilon_{kl}, \tag{1}$$

$C_{ijkl}$, the so-called stiffness tensor, can be rewritten in Voigt notation as a symmetric second-order tensor with six different stress elements ($\sigma_{11}, \sigma_{22}, \sigma_{33}, \sigma_{23}, \sigma_{13}, \sigma_{12} \equiv \sigma_1, \sigma_2, \sigma_3, \sigma_4, \sigma_5, \sigma_6$) and six different strain elements ($\varepsilon_{11}, \varepsilon_{22}, \varepsilon_{33}, \varepsilon_{23}, \varepsilon_{13}, \varepsilon_{12} \equiv \varepsilon_1, \varepsilon_2, \varepsilon_3, \varepsilon_4, \varepsilon_5, \varepsilon_6$), such that $ij \Rightarrow \alpha$ and $kl \Rightarrow \beta$:

$$C_{ijkl} = C_{\alpha\beta} = \begin{bmatrix} C_{11} & C_{12} & C_{13} & C_{14} & C_{15} & C_{16} \\ C_{21} & C_{22} & C_{23} & C_{24} & C_{25} & C_{26} \\ C_{31} & C_{32} & C_{33} & C_{34} & C_{35} & C_{36} \\ C_{41} & C_{42} & C_{43} & C_{44} & C_{45} & C_{46} \\ C_{51} & C_{52} & C_{53} & C_{54} & C_{55} & C_{56} \\ C_{61} & C_{62} & C_{63} & C_{64} & C_{65} & C_{66} \end{bmatrix} \tag{2}$$

This symmetric matrix has 21 different elements. For an isotropic material, there are only two independent elements:

$$C_{\alpha\beta} = \begin{bmatrix} K+4G/3 & K-2G/3 & K-2G/3 & 0 & 0 & 0 \\ K-2G/3 & K+4G/3 & K-2G/3 & 0 & 0 & 0 \\ K-2G/3 & K-2G/3 & K+4G/3 & 0 & 0 & 0 \\ 0 & 0 & 0 & G & 0 & 0 \\ 0 & 0 & 0 & 0 & G & 0 \\ 0 & 0 & 0 & 0 & 0 & G \end{bmatrix} \tag{3}$$

Therefore, just shear and bulk modulus or any other two elastic constants such as Young modulus and Poisson's ratio, which is normally estimated from standard UCS tests, are required to model rock deformation. For anisotropic materials, the stiffness matrix needs more elastic constants. For instance, minerals formed in a cubic crystal system, which is the simplest anisotropic crystalline form, the stiffness matrix has three different independent elements including $C_{11}$, $C_{12}$ and $C_{44}$:

$$C_{\alpha\beta} = \begin{bmatrix} C_{11} & C_{12} & C_{12} & 0 & 0 & 0 \\ C_{12} & C_{11} & C_{12} & 0 & 0 & 0 \\ C_{12} & C_{12} & C_{11} & 0 & 0 & 0 \\ 0 & 0 & 0 & C_{44} & 0 & 0 \\ 0 & 0 & 0 & 0 & C_{44} & 0 \\ 0 & 0 & 0 & 0 & 0 & C_{44} \end{bmatrix} \tag{4}$$

The studied igneous rocks are contained minerals that are formed in different crystalline systems. Different crystalline systems have different independent elastic elements; a triclinic system, for instance, has 21 independent elements. More details regarding the elastic properties of individual minerals can be found in [7]. In the presented methodology, igneous rocks are considered isotropic materials. It is notable that according to the petrographic analysis of the studied rocks reported in [5,10], minerals in igneous rocks do not show orientation, and these rocks can be considered isotropic. Then, the experimentally determined shear and bulk moduli of each mineral constituent are used to quantify the effective shear and bulk moduli of the studied rocks that can be used to determine other required elastic constants (Equation (3)).

### 2.3. Voigt–Reuss–Hill Average

Voigt–Reuss–Hill average [9] can be utilized to determine the effective mechanical properties of a heterogeneous polycrystalline solid [14,15]. Voigt–Reuss–Hill is the arithmetic average of the Voigt upper bound (isostrain) and Reuss lower bound (isostress) effective elastic moduli, presented in Equations (5) and (6), respectively:

$$M_V = \sum_{i=1}^{N} f_i M_i, \tag{5}$$

$$\frac{1}{M_R} = \sum_{i=1}^{N} \frac{f_i}{M_i}, \tag{6}$$

where $f_i$ and $M_i$ are the volume fraction and the elastic moduli of the ith component of a mixture of $N$ material phases (minerals and pores in rocks). By assuming that an isotropic rock material is dealt with, the shear modulus ($G$) and bulk modulus ($K$) of the mineral phases of the rock material can be used to obtain its elastic properties. Computed bulk and shear effective moduli according to the Voigt–Reuss–Hill average for the studied rocks are presented in Table 3.

### 2.4. Effect of Porosity on Effective Moduli

The Voigt–Reuss–Hill average has little practical value, except in the case where the constituent endmembers are elastically similar, as with a mixture of minerals without pore space [8]. The Voigt and Reuss bounds could be applied to a porous rock by treating the pore space as an additional component that has $K_i = G_i = 0$. However, the resulting Reuss bound will be zero, so other methods must be used to account for the effect of voids [3]. In this study, dry rock is considered a two-phase material in which the solid part of the rock is considered as the first phase, based on the effective medium theory, and the assumption that the solid part of a dry rock is homogenous. For this part, as proved by Brace [16], the Voigt–Reuss–Hill average would be a robust method to compute effective bulk and shear moduli. Then, pores are considered as the second phase. There are some models for considering two phase materials including matrix and inclusion as homogenous materials, among which Hashin–Shtrikman bounds [17] and Kuster and Toksoz formulation for effective moduli [18] seem to be more appropriate for estimating elastic moduli of dry rocks containing pores.

**Table 3.** Bulk and shear moduli of the solid phase of the studied rocks calculated using the Voigt–Reuss–Hill average.

| Rock Code | K (GPa) | | | G (GPa) | | |
|---|---|---|---|---|---|---|
| | Voigt | Reuss | Hill | Voigt | Reuss | Hill |
| R1 | 55.0 | 51.7 | 53.3 | 34.7 | 33.5 | 34.1 |
| R2 | 52.6 | 49.3 | 50.9 | 36.3 | 35.0 | 35.7 |
| R3 | 53.6 | 50.7 | 52.2 | 35.1 | 33.9 | 34.5 |
| R4 | 74.9 | 70.4 | 72.6 | 41.8 | 38.5 | 40.2 |
| R5 | 56.4 | 52.1 | 54.2 | 36.7 | 34.9 | 35.8 |
| R6 | 95.4 | 88.7 | 92.0 | 55.6 | 46.7 | 51.1 |
| R7 | 52.9 | 49.5 | 51.2 | 35.6 | 34.2 | 34.9 |
| R8 | 52.2 | 49.1 | 50.6 | 35.5 | 34.0 | 34.7 |
| R9 | 51.3 | 48.2 | 49.8 | 35.9 | 34.3 | 35.1 |
| R10 | 54.7 | 51.6 | 53.2 | 34.1 | 32.6 | 33.4 |
| R11 | 55.6 | 49.1 | 52.4 | 36.2 | 33.8 | 35.0 |
| R12 | 59.4 | 54.0 | 56.7 | 37.2 | 35.1 | 36.2 |
| R13 | 69.1 | 63.5 | 66.3 | 37.9 | 35.2 | 36.5 |
| R14 | 74.2 | 67.7 | 70.9 | 41.4 | 37.0 | 39.2 |
| R15 | 78.1 | 74.7 | 76.4 | 40.4 | 37.1 | 38.8 |
| R16 | 75.6 | 71.7 | 73.6 | 39.4 | 35.9 | 37.7 |
| R17 | 55.6 | 52.2 | 53.9 | 34.4 | 33.2 | 33.8 |
| R18 | 93.0 | 84.9 | 88.9 | 52.5 | 44.5 | 48.5 |
| R19 | 93.2 | 86.8 | 90.0 | 50.9 | 45.6 | 48.2 |
| R20 | 85.9 | 79.7 | 82.8 | 46.5 | 40.7 | 43.6 |
| R21 | 84.6 | 78.8 | 81.7 | 45.5 | 40.1 | 42.8 |
| R22 | 64.1 | 59.4 | 61.7 | 36.6 | 33.8 | 35.2 |
| R23 | 64.3 | 61.1 | 62.7 | 34.6 | 32.7 | 33.6 |
| R24 | 60.4 | 57.0 | 58.7 | 34.1 | 32.0 | 33.1 |
| R25 | 72.6 | 69.5 | 71.1 | 37.9 | 35.6 | 36.8 |
| R26 | 60.0 | 54.7 | 57.4 | 37.4 | 35.2 | 36.3 |
| R27 | 61.3 | 55.2 | 58.2 | 38.1 | 36.1 | 37.1 |
| R28 | 78.6 | 74.7 | 76.6 | 42.1 | 38.3 | 40.2 |

*2.5. Kuster and Toksoz Formulation*

Kuster and Toksoz [18] derived expressions for P- and S-wave velocities by using a long-wavelength first-order scattering theory. This expression for the bulk effective modulus $K_{KT}^*$ and shear effective modulus $G_{KT}^*$ can be written as [19]:

$$(K_{KT}^* - K_m)\frac{\left(K_m + \frac{4}{3}G_m\right)}{\left(K_{KT}^* + \frac{4}{3}G_m\right)} = \sum_{i=1}^{N} x_i(K_i - K_m)P^{mi} \qquad (7)$$

$$(G_{KT}^* - G_m)\frac{(G_m + \zeta_m)}{(G_{KT}^* + \zeta_m)} = \sum_{i=1}^{N} x_i(G_i - G_m)Q^{mi}, \qquad (8)$$

where the coefficients $P^{mi}$ and $Q^{mi}$ describe the effect of inclusion of material *i* in a background medium m. These coefficients can be calculated for specific shapes. In this study, it is considered that only one spherical pore space is within a medium of minerals as the background, such that *N* and $x_i$ are equal to 1. Notably, this should be considered as the representative elementary volume (REV) of the studied rocks. Therefore, Equations (7) and (8) can be calculated for the studied rocks by replacing $K_m$ and $G_m$ with calculated effective bulk and shear moduli using the Voigt–Reuss–Hill average; replacing $K_i$ and $G_i$ with bulk and shear moduli of the air, which are 0.101 MPa and zero, respectively; and by calculating $P^{mi}$ and $Q^{mi}$ using the following formulas [20]:

$$P^{mi} = \frac{K_m + \frac{4}{3}G_m}{K_i + \frac{4}{3}G_m} \qquad (9)$$

$$Q^{\text{mi}} = \frac{G_m + \zeta_m}{G_i + \zeta_m} \tag{10}$$

where:

$$\zeta = \frac{G}{6} \frac{(9K + 8G)}{(K + 2G)} \tag{11}$$

## 3. Results and Discussions

In this study, a new method is suggested for determining elastic constants of igneous rock materials by determining their porosity and modal composition and employing the effective medium theory. This method avoids destructive tests and the usage of energy as well as time-consuming and expensive tests.

In summary, the bulk and shear moduli of the solid phase of the studied rocks are calculated using the Voigt–Reuss–Hill average (Table 3). Then, using the Kuster and Toksoz formulation, the effect of porosity or the pore inclusions has been taken into account for calculating the effective moduli of the studied igneous rocks. The calculated effective moduli of the studied rocks following this method are presented in Table 4.

**Table 4.** Porosity, computed effective shear modulus and computed effective bulk modulus of the studied rocks.

| Rock Code | Porosity | *K* (GPa) | *G* (GPa) |
|---|---|---|---|
| R1 | 1.13 | 52.01 | 33.35 |
| R2 | 1.01 | 49.85 | 34.99 |
| R3 | 1.25 | 50.83 | 33.66 |
| R4 | 0.65 | 71.50 | 39.70 |
| R5 | 1.4 | 52.61 | 34.82 |
| R6 | 0.89 | 90.10 | 50.22 |
| R7 | 0.87 | 50.27 | 34.30 |
| R8 | 0.86 | 49.70 | 34.11 |
| R9 | 0.84 | 48.94 | 34.52 |
| R10 | 0.76 | 52.32 | 32.90 |
| R11 | 1.91 | 50.32 | 33.70 |
| R12 | 1.49 | 54.89 | 35.15 |
| R13 | 2.76 | 62.13 | 34.59 |
| R14 | 0.57 | 69.96 | 38.77 |
| R15 | 0.67 | 75.14 | 38.30 |
| R16 | 1.26 | 71.36 | 36.80 |
| R17 | 1.31 | 52.37 | 32.94 |
| R18 | 0.55 | 87.75 | 47.99 |
| R19 | 0.18 | 89.61 | 48.03 |
| R20 | 0.18 | 82.44 | 43.45 |
| R21 | 0.26 | 81.19 | 42.59 |
| R22 | 2.31 | 58.50 | 33.65 |
| R23 | 1.46 | 60.55 | 32.66 |
| R24 | 3.57 | 54.04 | 30.88 |
| R25 | 0.74 | 69.83 | 36.28 |
| R26 | 1.07 | 56.07 | 35.54 |
| R27 | 1.19 | 56.71 | 36.24 |
| R28 | 0.29 | 76.06 | 39.98 |

Based on these results, the bulk modulus of igneous rocks range between 48 and 91 GPa, while their shear modulus is between 30 and 51 GPa, and acidic igneous rocks show a lower modulus in comparison to the basic ones. These results confirm the experimental studies on determining elastic constants of igneous rocks [21].

Finally, it is notable that the method that is suggested in this study needs further verification by means of either numerical techniques or well-designed experimental methodologies.

## 4. Conclusions

In this study, a method is suggested for determining elastic constants of igneous rock materials, which requires the determination of the porosity and modal composition of the rock materials. Accordingly, based on modal composition, effective elastic constants of the studied igneous rocks are determined using the Voigt–Reuss–Hill average and elastic moduli of different mineral constituents. Then, the studied rocks are considered as a two-phase material, and the Kuster–Toksoz formulation is used to take into account the effect of porosity. The estimated elastic constants following this methodology are in good agreement with the experimental measurement of elastic constants of similar rock types. This method avoids destructive tests and the usage of energy for performing time-consuming and expensive tests and requires simple equipment.

**Author Contributions:** Conceptualization, S.A. and M.K.; methodology S.A. and M.K.; software, S.A.; validation, S.A. and M.K.; formal analysis, S.A.; investigation, S.A. and M.K.; resources, S.A. and M.K.; data curation, S.A. and M.K.; writing—original draft preparation, S.A. and M.K.; writing—review and editing, S.A. and M.K. All authors have read and agreed to the published version of the manuscript.

**Funding:** This research received no external funding.

**Data Availability Statement:** Data can be accessed by contacting the corresponding author.

**Conflicts of Interest:** The authors declare no conflict of interest.

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
