# Peer review of "Computing Elastic Moduli of Igneous Rocks Using Modal Composition and Effective Medium Theory"

_geosciences, doi:10.3390/geosciences12110413_

Round 1
Reviewer 1 Report
In this paper, a method is suggested for determining elastic constants of rock materials by determining their porosity and modal composition, and employing effective medium theory. If this method is reliable and effective, it does avoid destructive tests, usage of energy, time consuming and expensive tests. However, the reliability and validity of this method are lacking, especially the comparison with the experimental results.
This paper gives a new calculation method, but we can't judge whether the calculation results are correct or not.
In fact, if the porosity is treated as a component, we can also obtain elastic moduli using the Voigt–Reuss–Hill average model. But which calculation results are closer to the truth? If the latter are, the new method in this paper has no scientific significance.
So I hope the author can give a comparison with the experimental results.
Author Response
Dear Reviewer
Thanks for your consideration of our work. Your comments were insightful and helped authors improve the manuscript. The authors considered your views and revised the manuscript accordingly.
It is notable determining experimental determination of elastic moduli is a complicated tasks and needs well-designed experiments that is out of scope of this study. However, in future work it should be considered.
Reviewer 2 Report
This paper presents the results of computing elastic moduli using modal composition and effective medium theory in igneous rocks. For the remarks made during the revision of the document, I recommend reconsidering after a major revision (view the document attached).

Author Response
Dear Reviewer
Thanks for your consideration of our work. Your comments were insightful and helped authors improve the manuscript. The authors considered your views and revised the manuscript accordingly.
Round 2
Reviewer 1 Report
I appreciate that the author has made some modifications. However, the validation of the reliability of the new methods is still missing.
Author Response
Dear reviewer,
Thank you very much for considering our work. Regarding the experimental validation it is out of scope of this study. However, we have added some information regarding similar experiments that follow our results. It will be definitely considered in our future work
Reviewer 2 Report
The suggested considerations were corrected.
Author Response
Dear reviewer,
Thank you very much for considering our work.